# Targeting CXCR4 and CD47 Receptors: An Overview of New and Old Molecules for a Biological Personalized Anticancer Therapy

**DOI:** 10.3390/ijms232012499

**Published:** 2022-10-18

**Authors:** Manuela Leo, Lina Sabatino

**Affiliations:** Department of Sciences and Technologies, University of Sannio, Via Francesco de Sanctis, 82100 Benevento, Italy

**Keywords:** CXCR4, CD47, CXCL12, tumor cell proliferation, immune escape, antagonists, anticancer therapy, immunogenic surrender

## Abstract

Biological therapy, with its multifaceted applications, has revolutionized the treatment of tumors, mainly due to its ability to exclusively target cancer cells and reduce the adverse effects on normal tissues. This review focuses on the therapies targeting the CXCR4 and CD47 receptors. We surveyed the results of early clinical trials testing compounds classified as nonpeptides, small peptides, CXCR4 antagonists or specific antibodies whose activity reduces or completely blocks the intracellular signaling pathways and cell proliferation. We then examined antibodies and fusion proteins against CD47, the receptor that acts as a “do not eat me” signal to phagocytes escaping immune surveillance. Despite these molecules being tested in early clinical trials, some drawbacks are emerging that impair their use in practice. Finally, we examined the ImmunoGenic Surrender mechanism that involves crosstalk and co-internalization of CXCR4 and CD47 upon engagement of CXCR4 by ligands or other molecules. The favorable effect of such compounds is dual as CD47 surface reduction impact on the immune response adds to the block of CXCR4 proliferative potential. These results suggest that a combination of different therapeutic approaches has more beneficial effects on patients’ survival and may pave the way for new accomplishments in personalized anticancer therapy.

## 1. Introduction

One of the most challenging tasks in cancer biology is the search for new and more effective treatments. The introduction of biological molecules, the so-called biological therapy, has been a breakthrough in the treatment of many cancers with respect to classical chemotherapy as it slows down tumor growth, prevents metastasis formation and spreading, and reduces, in many cases, the amount and intensity of the adverse side effects. The spectrum of this innovative approach is expanding every day due to new achievements which improve the design and the number of the molecules, their efficacy and specificity of action, as well as the chance to be used in combination with other compounds [1].

Biological cancer therapy perfectly fits into the emerging concept of precision medicine and personalized cancer medicine. Accordingly, treatments should be tailored to each single patient based on (1) Accurate collection of clinical data; (2) Acquisition of imaging and laboratory data; (3) Results obtained by next-generation sequencing (NGS) technologies capable of detecting new, rare mutations or gene copy number variations in cancer cells as well as epigenetic modifications. Biological cancer therapy specifically involves treatments with natural or synthetic molecules which can attack tumor cells directly or indirectly by supporting and improving the immune system to fight cancer. These include monoclonal antibodies, adoptive cell transfer, gene therapy, treatment with cytokines, cancer vaccines, oncolytic viruses, immunoconjugates and the use of targeted therapy. In such a therapeutic approach, molecules that are directed towards genetic aberrations in oncogenes and tumor suppressor genes are essential [2]. 

CXCR4 and CD47 are two surface receptors whose role in tumorigenesis is well-known. Against these receptors, numerous types of molecules with distinct chemical structures have been used as inhibitors or antagonists; their number is increasing, and they are systematically scrutinized and susceptible to further development. 

CXCR4 is a G-protein coupled receptor (GPCR) expressed ubiquitously during embryogenesis and involved in important developmental processes. CXCR4 affects tumor growth; its overexpression is a driver of a large number of human malignancies and a marker of poor prognosis, especially in patients with breast, prostate, colorectal and lung cancer [3]. C-X-C motif chemokine 12 (CXCL12, also known as stromal cell-derived factor 1, SDF-1) is the chemokine ligand for CXCR4. The specific interaction activates intracellular signaling pathways that stimulate cell proliferation and support a feed-forward regulation loop that further enhances tumor progression [4].

CD47 also is a membrane receptor expressed in all normal cells [5,6]. The main ligand to CD47 is Signal-Regulatory Protein α (SIRPα), expressed on the surface of immune cells. This cross-interaction activates a downstream cascade leading to the inhibition of macrophage phagocytosis [7,8,9]. CD47, in fact, marks the cells as “self” and with a “do not eat me” signal, so that cells overexpressing CD47 are no longer phagocytosed; unfortunately, a variety of malignant cells overexpress CD47 escaping immune recognition and removal by phagocytosis [7]. 

Starting from these premises, in the last years, both CXCR4 and CD47 have been selectively targeted with different classes of molecules acting either as antagonists or inhibitors to block their intracellular signaling and proliferation or improve cellular recognition and immune response, respectively. 

This article will review and discuss some aspects of diverse CXCR4 and CD47 antagonists. Some of them have already been tested in early stages clinical trials, while some others can expectedly be used in single or combined treatments on the basis of our better understanding of cancer biology. For all, the expectation is to be used in more efficient treatments with beneficial effects on cancer patients’ survival. 

## 2. The CXCR4 Receptor

CXCR4 or C-X-C chemokine receptor type 4, also known as CD184, is a member of the seven-transmembrane, G-protein coupled receptor (GPCR) family, the largest class of cell surface receptors [3]. Human CXCR4 is a 352 amino acids long protein, shares 89% homology with the murine counterpart (359 amino acids), is ubiquitously expressed in both embryonic and adult tissues and regulates essential physiological processes, including embryogenesis, tissue repair, and hematopoiesis [10]. CXCR4 knockout mice display embryonic lethality due to widespread defects, altered vascularization of the gastrointestinal tract, generation of B lymphocytes and myeloid cells, formation of the cerebellum and heart defects [11]. CXCR4 conditional knockout mice disclose an even larger involvement during development in myogenesis, innervation of limbs and formation of renal vasculature [12,13]. In adult life, high expression of CXCR4 occurs in bone marrow (BM) and cells of the immune system, with lower levels in most other tissues and organs. Specifically, in differentiated immune cells, CXCR4 expression controls the homeostatic trafficking for immune surveillance and host defense. Of note, CXCR4 acts as a co-receptor for some strains of human immunodeficiency virus and, thus, is involved in HIV infection together with CCR5, both expressed on the T cells’ surface. Specifically, in the early stages of infection, the virus uses CXCR4 for viral entry; subsequently, it takes advantage of CXR4 to promote Acquired Immune Deficiency Syndrome (AIDS) progression [14]. Due to all these activities, it contributes to the pathogenesis of multiple diseases, including autoimmunity, atherosclerosis, and neurodegeneration [15,16,17].

CXCR4 plays relevant roles also in the tumorigenic process by directly acting at multiple steps; it is, in fact, overexpressed in many tumor cells affecting growth and invasion, suggesting that CXCR4 is a driver of human malignancies and a marker of poor prognosis [18]. More recently, it has been shown that CXCR4 is also upregulated in cancer stem cells (CSCs), or tumor-initiating cells, a population of malignant cells within a tumor able to self-renew and differentiate to produce the heterogeneous lineages of cancer cells that constitute the tumor [19]. Activation of the signaling pathways downstream to CXCR4 stimulates CSCs proliferation leading to an increase in their total number and activation of functions, including self-renewal, local invasion and dissemination to distant organs and tissues ([3] and references therein]). This latter property has been specifically related to CSCs as they are the only subpopulation bearing stem cell properties and thus capable of generating metastases. In pancreatic cancers, these events have been shown to be mediated by a specific subpopulation of CSCs characterized by CD133 expression, a stemness marker, and CXCR4 overexpression with respect to parenchymal tumor cells; they are localized at the invasive front of the mass, suggesting a role in tumor dissemination due to the expression of CXCR4 and the responsiveness to the specific ligand CXCL12 produced by other cells present in the microenvironment. Similar events have also been reported in colorectal cancer (CRC) [20]. In the case of breast cancer (BC), the most frequent sites of metastasis are the lung, brain and bone because they are enriched in CXCL12-expressing fibroblasts. Binding to CXCR4 turns these cells into carcinoma-associated fibroblasts (CAFs), which start to secrete CXCL12, which, in turn, acts on CSCs and tumor cells, further enhancing proliferation and/or the metastatic potential towards novel sites [21,22]. Thus, these signals are driven in part by the chemokines emitted from cancer cells, generating a feed-forward loop that directly enhances the survival and invasion of malignant cells [23]. These events also highlight the tight interconnections between cancer and stromal cells to generate the so-called tumor microenvironment (TME), which ultimately establishes tumor progression via direct and indirect effects [24]. Moreover, hypoxia, a condition that occurs in both primary and metastatic tumors, drives transcription of CXCL12 and CXCR4 to further stimulate growth and metastasis via promoting the angiogenetic potential of molecules such as vascular endothelial growth factor (VEGF) [25,26,27]. Finally, expression and signaling of the CXCR4/CXCL12 axis in both immunosuppressive and effector immune cells regulate the balance of pro- and antitumor leukocytes recruited to a tumor [28]. CSCs have also been implicated in the acquisition of resistance to standard chemotherapeutic drugs and radiation, through various mechanisms that allow them to survive and make them the leading cause of treatment failure and recurrent cancers [20,29]. 

Overall, the findings reported here highlight the multiple functions of CXCR4 in tumor environments and support the association of CXCR4 overexpression with poor clinical data and a dismal prognosis in a variety of malignancies. Importantly, these data strongly support the possibility of targeting CXCR4 on CSCs and tumor cells not only to reduce the tumor mass but also to impair tumor regeneration and possibly resistance to therapy, improving the success of cancer therapy by using both challenging and opportunistic strategies.

## 3. Binding of CXCL12 to CXCR4

The interaction of CXCR4 with its ligand CXCL12 is followed by phosphorylation at multiple sites of its intracellular C-terminal domain that leads to the dissociation of the Gβγ and Gα subunits and activation of the RAS-MAPK, PI3K-AKT-mTOR, Jak2/3-STAT2/4, phospholipase C, NF-κB and JNK/p38 signaling [30]. These pathways lead to intracellular calcium mobilization and cell proliferation, migration, and survival; interestingly, cell proliferation and migration are mutually exclusive processes; yet, in this case, they are activated by the same signaling. Interaction of CXCL12 with CXCR4 also induces the recruitment of β-arrestins, which in turn signal via the MAPK p38 [30,31] and leads to clathrin-dependent internalization of the receptor-ligand complex [32] (Figure 1).

CXCR4 forms homodimers that are confined within lipid rafts [33]; ligand binding increases CXCR4 oligomers that internalize more efficiently than monomers and regulate intracellular signaling and subsequent desensitization [34,35,36]. CXCR4 forms higher-order homo-oligomers and heterodimers with other receptors providing a further layer of regulation of its dynamics and signaling. CXCR4 can, in fact, heterodimerize with CCR2, negatively affecting the binding of ligands and antagonists on itself and the interacting receptor [37,38]. CXCR4 also associates with CCR5 to recruit monocytes, memory T helper cells and eosinophils. CCR5 is, in fact, the receptor for CCL5/RANTES, a chemokine produced by T cells, macrophages and activated platelets.

On the surface of T cells, CXCR4 can form heterodimers also with CCR7 and binding of CCR7 ligands enhances CCR7 homo- and CXCR4/CCR7 heterodimerization, without affecting CCR7 expression levels [39]. On the same T lymphocytes, CXCR4 can heterodimerize with the T cell receptor (the TCR/CD3 complex) upon specific activation, stimulating the reciprocal intracellular signaling, ultimately affecting new T cell epitope formation and recognition as well as cytokine secretion and chemotaxis [40,41]. CXCR4 downregulation in T cells is associated with alteration of the structure of the immunological synapse, reducing T cell/Antigen Presenting Cells contacts [42,43]. Interestingly, CXCR4 can also form heterodimers with the B Cell Receptor (BCR) in B cells. Of the two BCRs expressed in mature B cells, CXCR4 interacts only with the one binding IgD and activates the downstream pathway in response to CXCL12 [44]. 

The plasticity of CXCR4 in forming homo- and/or heterodimers is particularly evident in tumor microenvironments where cells are exposed to multiple ligands and inhibitors, enhancing intracellular signaling. This suggests that targeting multiple receptors and signaling pathways with a single drug may pave the way for more successful therapeutic approaches.

## 4. The CXCR7 Receptor 

CXCL12 can bind another receptor, CXCR7 (C-X-C motif receptor 7), that belongs to the atypical chemokine family and is renamed ACKR3 (Atypical Chemokine Receptor 3). Although CXCL12 shows a 10-fold higher affinity for CXCR7 than CXCR4, the binding of CXCL12 to CXCR4 is kinetically favored because both association and dissociation rates of CXCL12 with CXCR7 are slower than CXCR4 [45]. CXCR7 is highly implicated in embryonic heart formation and development, as demonstrated in *Cxcr7* knockout mice (*Cxcr*7^−/−^), which die at birth due to abnormal heart valve development [46]. In the adult, CXCR7/ACKR3 is expressed in mesenchymal stromal cells, central nervous system cells (astrocytes, glial and neuronal cells), vascular smooth muscle cells and, among the immune cells, in not mature B cells, natural killer (NK) cells, basophils, dendritic cells (DCs) and CD4 but not CD8 T cells [47]. The phenotypic differences described for *Cxcr*4^−/−^ and *Cxcr*7^−/−^ mice [48], along with recent studies in zebrafish [49], support the hypothesis that CXCR7 and CXCR4 have specific and distinct biological roles. CXCR7 acts as a “scavenger” or “decoy” for extracellular CXCL12 and CXCL11, promoting constant internalization and recycling of the receptor and establishing a CXCL12 gradient. This is consistent with the role of CXCR7 in controlling chemokine concentrations in the extracellular space and limiting signaling via other receptors [50,51]. 

The binding of CXCL12 to CXCR7 activates the intracellular signaling by itself through the β-arrestins pathway [50]. The binding also induces phosphorylation of the receptor that protects it against degradation, preserving the CXCR7 scavenger function [52]. In contrast to CXCR4, CXCR7 internalization occurs even in the absence of ligand binding and does not lead to receptor degradation [46]. CXCR7 can form heterodimers with CXCR4 but also with other chemokine receptors such as CCR2, CCR7, CCR5, and CXCR3, modulating the CXCR4-activated calcium signaling [53,54,55].

## 5. CXCR4 Antagonists

The activity of the CXCR4 receptor can be blocked or hindered by a series of molecules acting as antagonists or inhibitors that can be grouped in three classes [56]: (1)nonpeptide CXCR4 antagonists;(2)small-peptide CXCR4 antagonists;(3)antibodies to CXCR4.

### 5.1. Nonpeptide CXCR4 Antagonists

The group of nonpeptide CXCR4 antagonists includes molecules such as AMD3100, AMD070 and KRH-1636. 

AMD3100 is a bicyclam compound in which two cyclam rings are linked through an aromatic bridge; it acts as a specific CXCR4 antagonist inhibiting CXCL12-induced chemotaxis and GTP-binding and does not cross-react with other chemokine receptors [57,58]. AMD3100, as well as other molecules of this group, was first used for the treatment of HIV infection. For these reasons, great efforts have been made aiming at identifying molecules able to block this interaction. During Phase I/II clinical trials, HIV patients and volunteers exhibited leukocytosis due to the mobilization of diverse hematopoietic cells from the BM. For this reason and the relatively limited effect on the viral load, AMD3100 was discontinued for HIV treatment. Yet, it continued to be studied as a mobilizing agent of hematopoietic stem cells (HSCs) [59,60,61], so in 2008 the US Food and Drug Administration (FDA) approved AMD3100 as Plerixafor for the treatment of patients with hematological diseases. Since then, numerous clinical trials have been carried out either as monotherapy or in combination with other drugs, especially for untreated or relapsed/refractory acute myeloid leukemia (r/r AML) (NCT01455025); non-Hodgkin’s lymphoma and multiple myeloma (MM) (NCT00322842, using AMD3100 + granulocyte colony-stimulating factor (G-CSF)) [62]; r/r MM (NCT00903968, using AMD3100 + Bortezomib) [63].

The published data from these Phase 1/2 clinical trials suggest that AMD3100 is more efficient than monotherapy when administered in combination with chemotherapy, with an acceptable safety profile and ability to mobilize leukemic cells into the peripheral circulation, leading to encouraging remission rates [64].

AMD3100 has also been tested in two Phase 1 clinical trials (NCT02179970) for solid tumors such as colorectal, advanced pancreatic and ovarian cancer, aiming at better understanding the changes in the composition of immune cells in the TME. The results on the pharmacokinetics and safety profile suggested a recommended infusion rate for further studies but also led to the design of a Phase 2 trial combining AMD3100 with an immune-checkpoint inhibitor (ICI) in pancreatic cancer patients (NCT04177810, in progress). Furthermore, in combination with bevacizumab, an anti-VEGF monoclonal antibody, AMD3100 was tested in a Phase 1 study in patients with recurrent high-grade glioma (NCT01339039) showing that the combination was well tolerated, with a series of markers consistent with VEGF and CXCR4 inhibition [65]. In association with other chemotherapeutics, AMD3100 was also administered to patients with newly diagnosed high-grade glioma (NCT01977677); however, the patients’ cohort was too small to draw statistically significant conclusions, so the study was extended to more patients [66] and also to a Phase 2 study (NCT03746080) with the addition of whole brain irradiation.

AMD070 (mavorixafor, X4P-001) is an orally available small molecule CXCR4 antagonist able to form a hydrogen bond between its benzimidazole group and the Tyr45 residue of CXCR4 [67]. Pre-clinical studies have demonstrated the efficacy and safety of AMD070, and its 50% inhibition concentration was similar to AMD3100 in MT-4 cells. Administration of AMD070 to healthy volunteers was associated with the appearance of mild, reversible and not dose-dependent side effects such as headache, vague neurologic symptoms and gastrointestinal problems [68]. AMD070 was administered to patients with advanced melanoma as monotherapy and in combination with pembrolizumab, an anti-PD-1 (Programmed Death-1) antibody (NCT02823405). When used as a single agent, AMD070 showed a modulation of the immune cell profile in the TME and an increase of CD8^+^ T cell infiltration. AMD070 was also tested in 2 clinical studies of clear-cell renal cell carcinoma (RCC). In a Phase I study (NCT02667886), in combination with axitinib, a VEGF receptor inhibitor, it was well tolerated and showed evidence of clinical activity; in another study, AMD070, in combination with nivolumab, an anti-PD-1 antibody, showed potential antitumor activity with a better response and a manageable safety profile (NCT02923531) [69].

KRH-1636 is an orally available nonpeptide CXCR4 antagonist and, due to the fact that it is absorbed from the duodenum into the bloodstream, can be useful also for the treatment of HIV infection [70]. In fact, it displays a potent antiviral activity both in vitro and in vivo and inhibition of HIV-1 replication in MT-4 cells and in PBL-SCID mice. KRH-1636, as AMD3100, does not induce CXCR4 internalization. This compound and its derivatives contain the structural motif Arg-Arg-2-Nal, which is also shared by some peptides CXCR4 antagonists [71]. Because of its more favorable pharmacokinetic properties than similar small peptides, KRH-1636 was further developed to produce a Phase 1 candidate, KRH-3955, that, however, did not enter clinical trials [72].

### 5.2. Small-Peptide CXCR4 Antagonists

Specific small-peptides acting as CXCR4 antagonists were identified following the screening of peptides that naturally occur after HIV infection. Starting from natural peptides, the synthesis of hundreds of chemically modified compounds was carried out, bringing to T22 [73], T134 [74] and T140 [75]. The peptide T22 (|Tyr5,12, Lys7|-polyphemusin II) is polyphemusin-derived and shows strong inhibitory effects against CXCR4; it was one of the first modified peptides with a potent HIV response and the 50% inhibitory concentration useful to fight AIDS, similar to AZT (3′-azido-3′-deoxythymidine), initially used for the treatment of the human disease [76]. Interestingly, T22 also shows the antimicrobial activity as it can reduce bacterial growth in liquid culture and is less toxic than typical antimicrobial agents such as GWH1 in NIH3T3, HeLa and MRC-5 cells. This finding is important not only because microbial infections can occur after tumor diffusion, but also because they can contribute to tumor formation. T22 can then exert a dual role as a CXCR4 antagonist targeting the downstream pathways in tumors overexpressing this receptor and as an antimicrobial agent [77]. 

T140 is a shorter form of T22 (14-mer with respect to 18-mer peptide) with a strong anti-HIV effect but low biostability. For this reason, to improve T140 effectiveness, chemical derivatives were produced, such as 4F-benzoyl-TN14003 and 4F-benzoyl-TE14011 that have a p-Fluorobenzoyl group at the N-terminus. Beyond anti-HIV activity, the T140-derivates reduce the migration in vitro and the ability to form metastasis in vivo of numerous cell lines derived from solid tumors, MM or diverse types of leukemias [78]. 4F-benzoyl-TN14003 induces mobilization of HSCs and many different progenitors from the BM within a few hours post-treatment in a dose-dependent manner [79]. TN14003 had no effect on mobilizing NK cells and was found to efficiently synergize with G-CSF in its ability to mobilize white blood cells and progenitors. TN14003 was significantly more potent in mobilizing HSCs and progenitors into the blood than AMD3100, with or without G-CSF [79].

The synthetic peptide BL-8040, also known as motixafortide, is a high-affinity CXCR4-antagonist. In human AML and MM mouse xenografts, BL-8040, used as monotherapy, displayed prolonged occupancy and sustained inhibition of the receptor, resulting in the induction of apoptosis of AML cells and mobilization of progenitor cells. When combined with cytarabine in a Phase IIa clinical trial of patients with r/r AML (NCT01838395), this peptide induced a 38% complete response and significantly improved patients’ overall survival compared to the antineoplastic anti-metabolite cytarabine alone [80]. In another clinical trial (NCT03154827, study terminated due to lack of enrollment), the safety and efficacy of motixafortide were tested in combination with atezolizumab, a programmed death-ligand 1 (PD-L1) inhibitor, in the maintenance treatment of AML patients aged 60 years or older (BATTLE Study). In a Phase III clinical trial for MM (NCT03246529, GENESIS), it was reported that patients who received BL-8040 + G-CSF produced more blood cells than those treated with placebo + G-CSF [81]. As far as solid tumors, in clinical trials for pancreatic cancer, BL-8040 was administered in combination with pembrolizumab or with pembrolizumab and chemotherapy in chemo-resistant (NCT02826486 COMBAT) [82] or metastatic pancreatic cancer patients (NCT02907099) [83]. In both cases, overall survival was significantly increased, demonstrating that targeting CXCR4 and PD-1 at the same time is a promising antitumor therapy.

The synthetic cyclopeptide CXCR4 inhibitors, POL6326 (balixafortide), efficiently mobilized engrafted leukemia cells from their protective stromal microenvironment into the circulation in a murine leukemia model; it also displayed a strong synergy in combination with G-CSF, significantly reducing leukemia burden and prolonging animals’ survival [84]. In a Phase 2a study (NCT01105403) of MM patients, POL6326 was shown to be safe and well tolerated with an efficient mobilization of HSCs [85]. With regard to solid tumors, POL6326 has been tested in combination with eribulin, an inhibitor of microtubule dynamics, in a Phase 1 trial (NCT01837095) of patients with relapsed triple negative and hormone-refractory ER-positive metastatic BC [86]. The anti-cancer response was sufficiently good, better than eribulin monotherapy, and safe with only minor adverse effects, suggesting that the combination of these two molecules represents a new therapeutic approach for metastatic BC [86]. More recently, and based on the positive results obtained, POL6326 has been proposed as the first CXCR4 antagonist to enter a Phase 3 clinical trial for BC [72]. 

LY2510924 is a potent and selective CXCR4 antagonist with a cyclic peptide structure able to block the pathways activated by CXCL12 as cell migration, CXCL12-induced GTP binding and downstream signaling. LY2510924 is stable in in vivo experiments and inhibits tumor growth in human xenograft models developed with RCC, non-Hodgkin lymphoma, lung, and colon cancer cells that express functional CXCR4 [87]. In preclinical models of AML, LY2510924 durably blocked CXCR4 and inhibited CXCL12–induced chemotaxis and pro-survival signals of AML cells more effectively than plerixafor. The antileukemia effect was also demonstrated in mouse models of AML as monotherapy and in combination with chemotherapy [88]. LY2510924 is also being studied in a Phase 1b trial (NCT02652871) combined with anthracycline idarubicin and cytarabine in 36 adult patients with r/r AML [89]. 

In early clinical tests, LY2510924, however, did not show efficacy in solid tumors [72]. Positive results were instead obtained in a Phase 1a study (NCT02737072) in which patients with refractory solid tumors were treated with LY2510924 in combination with durvalumab, a monoclonal human antibody against PD-L1; it was demonstrated that LY2510924 is safe and the combination therapy gave results similar to monotherapy suggesting a role of CXCR4 inhibition in recruiting blood cells, specifically T cells, to the TME [90].

A new class of cyclic peptides, CXCR4 antagonists, was recently developed using a ligand-based approach. Briefly, a three-residue segment was identified at the N-terminus of CXCL12, similar but in inverse order, to an inhibitory chemokine secreted by a herpes virus. This motif was the core of a cyclic peptide library tested in vitro for its ability to block CXCR4 function. Among the 19 peptides assessed, three (Peptide R, I and S) limited lung metastasis formation in vivo in syngeneic mice injected with melanoma or osteosarcoma cells; they also inhibited primary tumor growth in xenografts of human renal cancer cells. All three peptides were able to mobilize hematopoietic precursors in a CXCR4-dependent manner, similar and more durable than AMD3100 [91]. Peptide R showed the strongest effects, especially in impairing lung metastases [92]. Mice injected with human colon cancer cells administered with 5-Fluoroacil or Oxaliplatin, in the presence of Peptide R, displayed a relative 4-fold smaller tumor volume than chemotherapeutics alone [93]. 

### 5.3. Antibodies to CXCR4

Specific antibodies against CXCR4 were produced with the aim of blocking the CXCL12/CXCR4 axis signaling. Due to the conformational heterogeneity of the receptor, as reported above, it is not easy to produce monoclonal antibodies [94]. The most used monoclonal antibody is 12G5 which shows antiviral activity [95,96], inhibits proliferation and adhesion of HL-60 cells [97], and reduces tumor growth and micro-metastasis in human osteosarcoma xenograft mice [98]. 

The fully humanized antibody LY2624587 was tested in a Phase 1 study in patients with advanced or metastatic cancers (NCT01139788). The trial was completed with no results published to date. 

Ulocuplumab (MDX-1338) is a fully human anti-CXCR4 antibody that binds to the ECL2 sequence of CXCR4 with antagonistic activity to the receptor; it also showed promising antitumor activity and pro-apoptotic potential in the treatment as monotherapy of chronic lymphocytic leukemia cells in vitro characterized by high expression of CXCR4 [99]. In a Phase I trial (NCT03225716), patients affected by Waldenström macroglobulinemia, a disease characterized by a mutation in *MYD88* and *CXCR4*, were administered with ulocuplumab and ibrutinib, an inhibitor of Bruton’s tyrosine kinase, and showed a 2-year progression-free survival in 90% of cohort’s members [100]. Furthermore, in a Phase 1b clinical trial for r/r MM (NCT01359657), ulocuplumab was administered in combination with lenalidomide and dexamethasone, or bortezomib and dexamethasone. Benefits after the administration were also observed in patients previously treated with immunomodulatory agents, suggesting the importance of targeting the CXCR4 axis in MM [101]. 

A Phase 1 dose escalation/expansion trial (NCT01120457) in patients with r/r AML or selected B-cell cancers determined the maximum tolerated dose of ulocuplumab and assessed the safety and tolerability of the antibody in combination with chemotherapy MEC (mitoxantrone, etoposide, cytarabine). Ulocuplumab was given in escalation during monotherapy or in combination with MEC in the first cycle. In the expansion phase, patients received ulocuplumab and MEC with overall Complete Remission with Incomplete Count Recovery (CR/Cri) at 51%, far more favorable than the response rate for MEC alone (24–28%) [102]. Ulocuplumab was also combined with nivolumab in a Phase 1/2 study in advanced or metastatic pancreatic and SCLC (NCT02472977); however, the study was terminated early because of lack of efficacy [72].

PF-06747143, a humanized anti-CXCR4 immunoglobulin G1 (IgG1) antibody, is used to fight hematological malignancies and also inhibit downstream pathways activated by the CXCL12/CXCR4 axis. This molecule promotes the mobilization of blood cells and induces tumor cell death thanks to its Fc constant region-mediated effector function. Finally, PF-06747143, either as a single agent or in combination with standard chemotherapy, was tested in a Phase 1 clinical trial of AML (NCT02954653 terminated due to a change in sponsor prioritization) [103].

The CXCR4 antagonists employed in the clinical trials surveyed here are summarized in Table 1.

## 6. The CD47 Receptor

CD47, Cluster of Differentiation 47, is a transmembrane receptor belonging to the immunoglobulin (Ig) superfamily, it is ubiquitously expressed on all normal cells but overexpressed on hematological and solid malignant cells [104,105,106,107]; it is a highly glycosylated protein consisting of an extracellular N-terminal IgV-like domain, five highly hydrophobic membrane-spanning segments and a short hydrophilic cytosolic C-terminus. The IgV-like domain contains two disulphide bonds, and an additional bridge is formed between the extracellular and one of the transmembrane regions. CD47 is also called Integrin Associated Protein (IAP) as it was discovered as a plasma membrane molecule that copurifies with the integrin αvβ3; in fact, via its IgV-like domain, it interacts in cis with multiple integrins, including αIIbβ3 and α2β1, present in the same membrane [108].

The main ligand for CD47 is Signal-Regulating Protein α (SIRPα) expressed on the surface of monocytes, macrophages, neutrophils and DCs [5,6,109]. SIRPα, also known as CD172a or SHPS-1, is a glycoprotein of the Ig superfamily containing an extracellular domain and intracellular immunoreceptor tyrosine-based inhibition motifs (ITIMs). The interaction of CD47 with SIRPα (and γ) occurs in trans, i.e., between two different cells, mediates cell-cell adhesion and is highly species-specific [5]. The structure of the CD47/SIRPα complex has been solved by high-resolution crystallography and shows that the two molecules form a 1:1 stoichiometric complex. The SIRPα ligand binding domain is tightly interdigitated with CD47 so that their interactions are mainly mediated by loops at the intracellular side. The CD47/SIRPα binding induces phosphorylation of the intracellular ITIMs, triggering a downstream cascade that leads to disruption of the cytoskeleton and, ultimately, to inhibition of macrophage phagocytosis [7,9] (Figure 2).

CD47 also interacts with thrombospondins (TSPs), a family of five glycoproteins able to regulate cell migration; this specific binding abrogates VEGFR2 signaling, affecting proliferation, differentiation and inflammation [110]. Interactions with both integrins and thrombospondins activate Gi protein signal transduction. Besides specific integrins, CD47 directly associates with the Fas receptor in T cells, promoting Fas-mediated apoptosis. CD47 does not appear to interact with other cytoplasmic proteins; rather, it allows its downstream signaling through lateral interactions with other receptors, such as CD14, in human resting but not LPS-stimulated monocytes [111]. 

Importantly, the CD47/SIRPα interaction tags cells as “self” and with a “do not eat me” signal; this means that, in normal conditions, cells expressing low CD47 levels are phagocytosed, a pivotal step in the maintenance of tissue integrity and homeostasis. Unfortunately, CD47 is overexpressed on the surface of a variety of malignant cells, and its binding to SIRPα on macrophages suppresses their phagocytic activities. For this reason, the complex is also called “macrophage immune checkpoint”; its disruption, thus, stimulatesmacrophages-mediated phagocytosis of tumor cells and can be employed as a tool for producing next-generation immunoregulatory drugs [112]. To this goal, monoclonal antibodies and chimeric proteins have been designed and tested to block the CD47/ SIRPα interaction for efficient phagocytosis resulting in tumor rejection and development of antitumor immunity [113]. Specific CD47/SIRPα targeting antibodies have shown attractive results against various cancers including AML [114], anaplastic thyroid carcinoma [115], lymphoma [107], BC [116]. 

In particular, magrolimab (Hu5F9-G4, ONO-7913) is a humanized monoclonal antibody that binds CD47 at low nanomolar affinity and is built on an IgG4 scaffold to minimize Fc-mediated effector toxicity for non-tumor cells expressing CD47 [117]. In immunodeficient mice models, magrolimab shows strong monotherapy activity against human hematological malignancies [118]. Many clinical trials are currently undergoing to evaluate its efficacy in several tumor types. 

In a Phase 1b trial of AML patients (NCT03248479), magrolimab was used in combination with azacitidine [119]; it was also employed in Phase 1 of advanced solid tumors (NCT02216409) [120]. Patients affected by myelodysplastic syndromes were treated with magrolimab and azacitidine (NCT03248479). The combination therapy was well tolerated, with promising efficacy. This trial was followed by NCT04313881, which is in progress [121].

Lemzoparlimab (TJC4) is also a humanized monoclonal antibody and, as magrolimab, contains an IgG4 scaffold and binds specific CD47 epitopes. Glycosylation near the binding epitopes on red blood cells (RBCs), “shields” them from antibody binding. Lemzoparlimab, in fact, had minimal and transient effects on RBCs in primates. However, despite this mechanism of binding, anemia was observed in 30% of patients treated with this antibody. In a Phase I study, TJC4 was used for r/r advanced solid tumors and lymphoma as a single agent or in association with pembrolizumab or rituximab (NCT03934814) [122].

AO-176 is an anti-CD47 antibody based on an IgG2 Fc domain that binds selectively to CD47 on tumor cells but not on other cells. AO-176 does not activate antibody-dependent cellular cytotoxicity but does have a direct cell-killing effect; it has minimal effect on hematologic parameters in primates with no anemia. However, data from Phase I/II trials of AO-176 (NCT03834948, NCT0445701) in multiple solid tumors showed that it induces phagocytosis of tumor cells, while preserving T cells and causes thrombocytopenia in 33% and anemia in 22% of the cases [123,124].

SRF231 is an anti-CD47 fully human IgG4 monoclonal antibody that binds CD47 on the surface of RBCs but does not cause phagocytosis [125]; it has been proposed in Phase 1 preclinical model of both advanced solid and hematologic malignancies. (NCT03512340) [126].

B6H12.2 is a commercially available anti-CD47 monoclonal antibody that efficiently blocks the interaction between CD47 and SIRPα in Laryngeal Squamous Cell Carcinoma (LSCC) in vitro. This neoplasm is characterized by overexpression of CD47, and it has been found that the use of B6H12.2, or other monoclonal antibodies, either promotes CD47 decrease or the stimulation of phagocytosis by macrophages, highlighting the importance of this receptor in the maintenance of this neoplasm [127].

CC-90002, also known as INBRX 103, is a humanized monoclonal antibody based on an IgG4 scaffold. In the clinical trial of patients with r/r AML or high-risk myelodysplastic syndromes (NCT02641002), treatment with CC-90002 monotherapy produced the expected anemia in 32% of patients; thrombocytopenia also occurred in 39% of patients, suggesting an unknown mechanism, independent of antibody-dependent cellular cytotoxicity and complement-dependent cytotoxicity activation [128].

Among fusion proteins, ALX148 (evorpacept) has been engineered to contain two CD47 binding domains of SIRPα linked to an inactive Fc region of a human immunoglobulin. ALX148 is being evaluated alone as monotherapy or in combination with other anticancer antibodies or with ICIs in both solid non-small cell lung carcinoma (NSCLC), head/neck squamous cell carcinoma (HNSCC), gastric/esophageal junction carcinoma and hematologic malignancies [129]. In a Phase I trial of patients with lymphoma or advanced or metastatic solid tumors with no available standard therapy (NCT03013218, ASPEN-01), ALX148 was well tolerated and able to contrast tumor growth when administered in combination with pembrolizumab for HNSCC and NSCLC, or trastuzumab for HER2-positive gastric or gastroesophageal junction cancers [130]. 

Trillium (TTI-621) is a fusion protein containing the N-terminal V-domain of human SIRPα and the IgG1 Fc domain. This “decoy receptor” can bind CD47 on tumor cells, and can activate phagocytosis and Fc effector functions for maximum efficacy. TTI-621 binds CD47 on a variety of hematologic cells and causes anemia in primates but exhibits minimal binding to human RBCs, presumably because it binds to CD47 in clusters in the cell membrane but not when it is distributed and associated with a cytoskeletal protein, spectrin, in human RBCs [131]. Trillium was tested in Phase 1 clinical trial of r/r hematologic malignancies (NCT02663518), either as a single agent or in combination with rituximab or nivolumab, and emerged that was well tolerated with good efficacy [132]. Only recently, Trillium has been tested in combination with doxorubicin in patients with unresectable or metastatic leiomyosarcoma (NCT04996004). 

Table 2 shows the anti-CD47 antibodies and fusion proteins employed in the clinical trials surveyed here.

## 7. CXCR4 and CD47 Interactions

All the data presented so far underscore the role that CXCR4 and CD47 play in tumor promotion. We recently disclosed an unexpected counterbalancing activity of the CXCL12/CXCR4 axis, i.e., promotion of cell phagocytosis by macrophages through lateral interaction with CD47. Indeed, CXCR4 and CD47 already physically interact in specialized membrane regions and the binding of BoxA, a fragment of HMGB1 corresponding to the first domain of the protein, triggers co-internalization of the complex, leading to reduced exposure on the cell surface. We have shown that BoxA also stimulates the emission of HMGB1 and ATP along with membrane exposure of Calreticulin, the so-called Damage Associated Molecular Patterns (DAMPs) [133,134]. Extracellular HMGB1 interacts with the Toll-Like Receptor 4 (TLR4) on the surface of adjacent immature DCs, promoting their maturation and release of chemokines and cytokines in the neighboring microenvironment along with CXCL12 that, in turn, triggers the release of more DAMPs [135]. Ecto-calreticulin interacts with CD91 on the surface of macrophages and DCs, acting as an “eat me” signal that favors the identification and ingestion of tumor cells by phagocytes. All these events reduce surface exposure of the “do not eat me” signal CD47 after stimulation by BoxA, make tumor cells more visible to macrophages and elicit a robust phagocytosis [133]. 

Interestingly, also the administration of CXCL12 to co-culture of mesothelioma cells with BM-isolated macrophages, leads to enhanced ingestion of tumor cells. This is not unexpected as HMGB1, from which BoxA is derived, has many cross-functions with CXCL12, which can form heterodimers and interact with CXCR4. The striking difference is that CXCL12 acts as a tumor promoter, while BoxA promotes tumor cell phagocytosis and tumor rejection, protecting mice from a rechallenge due to a robust immunological memory [133]. It is conceivable that phagocytosis of tumor cells is not sufficient to reduce the tumor mass and that antitumor effector T cell clones should be generated to recognize tumor-associated antigens and kill more tumor cells [133,134]. At the moment, it is not known whether additional and/or diverse signals conveyed by the microenvironment are required to distinguish between healthy and tumor cells [134]. We designated this process as “ImmunoGenic Surrender” (IGS), whereby engagement of CXCR4 induces co-internalization and surface depletion of CD47 in tumor cells, followed by phagocytosis by macrophages (Figure 3).

This ensures the production of new tumor antigens and tumor-specific T cell clones, which cause tumor rejection and protect from subsequent challenges for a long-lasting immune memory. IGS can be then envisaged as a survey process that exposes tumor tissues to immunosurveillance: in a small percentage of cases, this event leads to tumor rejection and antitumor immunity; in a large percentage, instead, the tumor-promoting activity of the CXCL12/CXCR4 axis prevails. Importantly, treatment with exogenous BoxA enhances IGS and leads to tumor rejection in a considerable fraction of mice. This underlines that other exogenous CXCR4 ligands, like BoxA, might be designed and employed as promising antitumor compounds in combination with anti-CD47 antibodies, as described above. The former, in fact, would provide “eat me” signals on tumor cells, while targeting CD47 would remove “do not eat me” signals with an IGS enhancement [133].

## 8. Conclusions and Perspectives

Despite the progress made, cancer is still considered a disease with an uncertain prognosis, and novel treatments are eagerly awaited. Biological therapy, with all its multifaceted applications and recent achievements, has provided a wealth of data on the efficacy of novel molecules that can expectedly be used in clinical practice with beneficial effects at the bedside. In this paper, we reviewed the current knowledge on selected molecules as nonpeptides, small-peptides and antibodies acting as antagonists to the CXCR4 receptor and antibodies and chimeric proteins directed against CD47, two well-established mediators of tumorigenic signals. Yet, the intracellular signaling activated by CXCR4 ligand binding has not been fully elucidated; moreover, lateral interactions with other pathways are still undefined, widening the scenario in which CXCR4-inhibiting compounds could act. We focused only on those compounds which have already shown therapeutic effects and on those displaying promising and favorable results in early-stage clinical trials.

These latter tests, especially those conducted in leukemias or blood diseases, have provided encouraging results and support the rationale for introducing CXCR4 inhibitors in clinical practice. On the other hand, the drawback of some pilot studies relative to other compounds of this class forced the search for novel drugs to be eventually tested. There is still a significant need for better treatment approaches with newly disclosed and/or synthesized compounds for the high proportion of patients whose disease relapses or is refractory to. Future clinical trials should also explore combinations of small molecules and/or CXCR4 inhibitors with other targeted therapies and schedules to optimize efficacy/safety benefits. Together, these preclinical and clinical studies strongly support CXCR4 inhibition as a promising antitumor therapeutic approach.

CD47 is overexpressed in cancer, and for this reason, it is the target of selected therapies. A series of diverse humanized monoclonal antibodies, as well as fusion proteins, have been synthesized and tested both in vitro and in numerous preclinical and clinical trials. The aim is to make tumor cells more visible to phagocytes for removal and to elicit a robust adaptive immune response by blocking the interaction with its ligand SIRPα.

The occurrence of hematological abnormalities such as anemia or thrombocytopenia, both in primates and humans, is a relevant drawback of these single treatments, dampening, in part, the enthusiasm for their use in the clinical practice. Attempts have, thus, been made to combine the anti-CD47 antibodies with other drugs to reduce the dosages and the undesired adverse effects, especially those on RBCs. Moreover, these associations attack the neoplasm from multiple points as the drugs influence diverse pathways with more significant results. These combinations should also be tested in future clinical trials with different therapeutic schedules to optimize efficacy/safety benefits. The association of selected anti-CD47 antibodies with immunotherapy is emerging as an important therapeutic approach, and encouraging results are obtained in early-stage clinical trials. Finally, the combination of treatments that block CXCR4 signaling and reduce CD47 cell exposure, in line with the newly discovered immunogenic surrender process, have been put forward in vitro and in animal models.

In conclusion, CXCR4 and CD47 can be targeted by specific molecules with beneficial effects on cancer patients’ survival. The design of novel and more efficacious molecules, the combination with other compounds directed to different receptors or cellular targets, may pave the way for a more successful personalized cancer therapy.

## Figures and Tables

**Figure 1 ijms-23-12499-f001:**
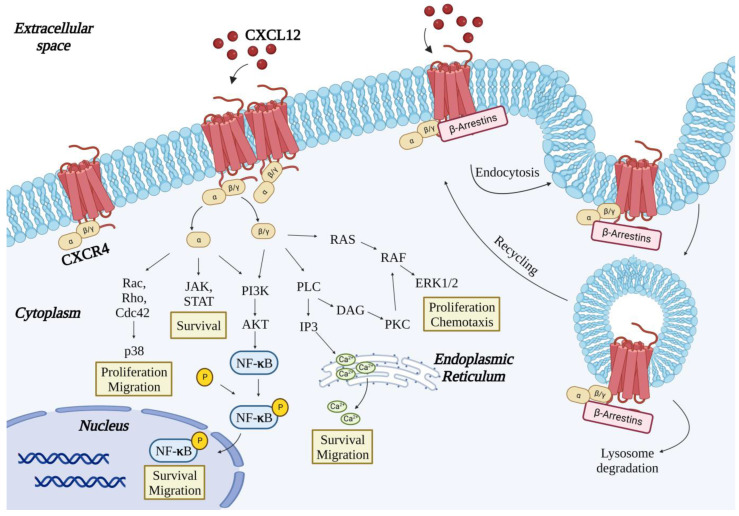
Schematic representation of the CXCR4 downstream signaling pathways activated by CXCL12 binding. The figure was created with BioRender.com (accessed on 7 October 2022).

**Figure 2 ijms-23-12499-f002:**
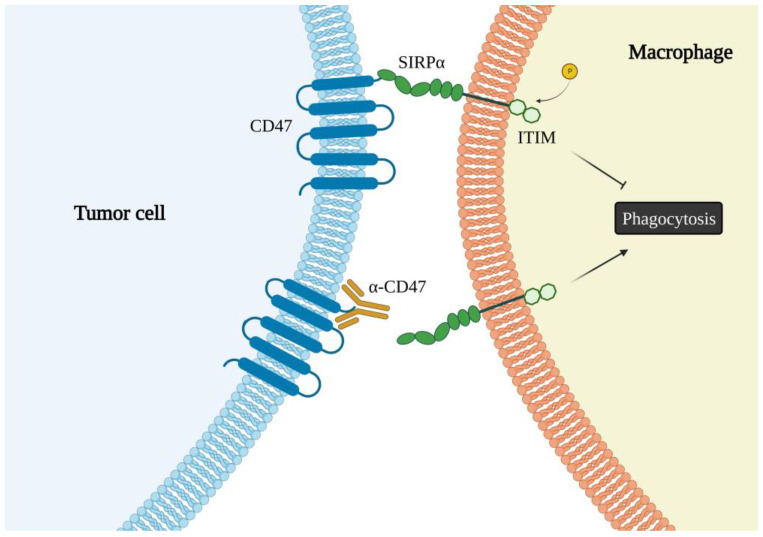
Schematic representation of the role of CD47 in phagocytosis. The binding of SIRPα to CD47 hampers phagocytosis of tumor cells by macrophages. Inhibition of SIRPα/CD47 interaction, as with a specific antibody, promotes engulfment of the cells and eradication of the tumor. The figure was created with BioRender.com (accessed on 8 October 2022).

**Figure 3 ijms-23-12499-f003:**
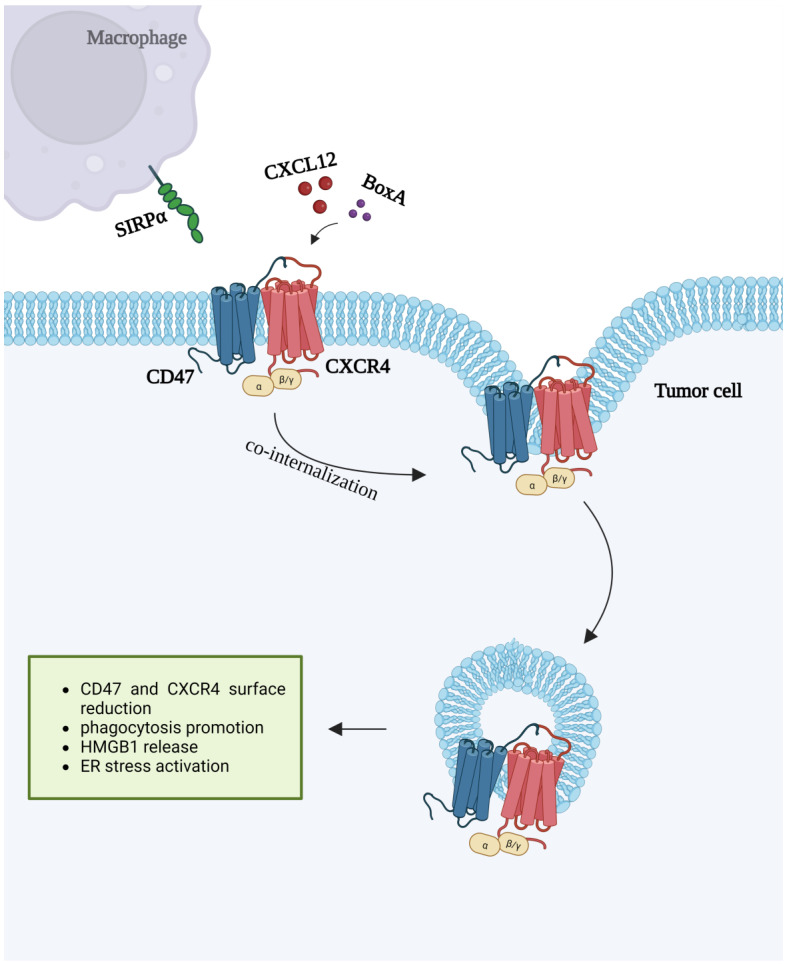
CXCL12 or BoxA binding induces co-internalization of CXCR4 and CD47 receptors. The figure was created with BioRender.com (accessed on 13 October 2022).

**Table 1 ijms-23-12499-t001:** List of the CXCR4 antagonists analyzed in this study.

Class	Name	Clinical Trial	Tumor	References
Nonpeptide	AMD3100, plerixafor	NCT01455025 completed	r/r AML	-
		NCT00322842 completed	non-Hodgkin’s lymphoma, MM	[62]
		NCT00903968 completed	r/r MM	[63]
		NCT02179970 completed	pancreatic, ovarian, CRC	-
		NCT04177810 in progress	pancreatic	-
		NCT01339039 terminated	high-grade glioma	[65]
		NCT01977677 completed	high-grade glioma	[66]
		NCT03746080 recruiting	glioblastoma	-
	AMD070, mavorixafor, X4P-001	NCT02823405 completed	advanced melanoma	-
		NCT02667886 active, not recruiting	RCC	-
		NCT02923531 completed	RCC	[69]
Small-peptide	BL-8040, motixafortide	NCT01838395 completed	r/r AML	[80]
		NCT03154827 terminated	AML	-
		NCT03246529 active, not recruiting	MM	[81]
		NCT02826486 active, not recruiting	metastatic pancreatic adenocarcinoma	[82]
		NCT02907099 active, not recruiting	metastatic pancreatic	[83]
	POL6326, balixafortide	NCT01837095 completed	metastatic BC	[86]
	LY2510924	NCT02652871 completed	r/r AML	[89]
		NCT02737072 terminated	solid tumor	[90]
Antibody	LY2624587	NCT01139788 completed	advanced or metastatic cancer	-
	MDX-1338, ulocuplumab	NCT03225716 active, not recruiting	Waldenström macroglobulinemia	[100]
		NCT01359657 completed	MM	[101]
		NCT01120457 completed	acute myelogeneus leukemia, B-cell cancers	-
		NCT02472977 terminated	solid tumor	-
	PF-06747143	NCT02954653 terminated	AML	[103]

**Table 2 ijms-23-12499-t002:** List of the CD47 inhibitors analyzed in this study.

Name	Clinical Trial	Tumor	References
Hu5F9-G4, ONO-7913, magrolimab	NCT03248479 active, not recruiting	hematologic, AML	[119]
	NCT02216409 completed	solid tumor	[120]
	NCT04313881 recruiting	myelodysplastic Syndromes	[121]
TJC4, lemzoparlimab	NCT03934814 active, not recruiting	advanced r/r solid tumor and lymphoma	[122]
AO-176	NCT03834948 active, not recruiting	multiple solid	[123,124]
SRF231	NCT03512340 completed	advanced solid tumor and hematologic	[126]
CC-90002, INBRX 103	NCT02641002 terminated	myelodysplastic and r/r AML	[128]
ALX148, evorpacept	NCT03013218 active, not recruiting	solid tumor and lymphoma	[130]
TTI-621, trillium	NCT04996004 recruiting	leiomyosarcoma	-
	NCT02663518 active, not recruiting	hematologic malignancies and solid tumor	[132]

## Data Availability

Not applicable.

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
