# Peer review of "Targeting CXCR4 and CD47 Receptors: An Overview of New and Old Molecules for a Biological Personalized Anticancer Therapy"

_ijms, 2022, doi:10.3390/ijms232012499_

Round 1

Reviewer 1 Report

The Authors review highly interesting topic, although, the lack of figures cause that the review is difficult to follow.

Author Response

Q.1 The Authors review highly interesting topic, although, the lack of figures cause that the review is difficult to follow.

Response: We thank the Reviewer for this suggestion. Accordingly, we added two cartoons that illustrate the signaling pathways activated by CXCR4 and CD47 receptors that are the topic of the article.

Reviewer 2 Report

In the manuscript # ijms-1924117 by Leo & Sabatino “Targeting CXCR4 and CD47 receptors: new and old molecules for a biological personalized anticancer therapy”, authors provide overview of the therapies targeting the CXCR4 and CD47 receptors. They summarize the results of early clinical trials testing compounds classified as non-peptide, small peptides CXCR4 antagonists or specific antibodies whose activity reduces or completely blocks the intracellular signaling pathways and cell proliferation. They have then focused their attention on selected biologics like antibodies and fusion proteins against CD47, the receptor that acts as a “do not eat me” signal to phagocytes escaping immune surveillance. Next, they discuss some drawbacks of those approaches that impair their use in the clinic. At the further parts of the manuscript, the authors discuss the “ImmunoGenic Surrender” mechanism that involves crosstalk and co-internalization of CXCR4 and CD47 upon engagement of CXCR4 by ligands or other (therapeutic) molecules. The favorable effect of such compounds is dual as CD47 surface reduction impact on the immune response adds to the block of CXCR4 proliferative potential. They favor the concept that combination of different therapeutic approaches has more beneficial effects on patients’ survival and may contribute to the success of the personalized anticancer therapy. This is potentially interesting, albeit relatively narrowly-focused review manuscript that needs substantial improvements.

Improvements suggestions:

1.)

While most of the manuscript is dedicated towards the immunotherapy, that show promise in eliminating not only the bulk of the tumor but also cancer stem cells – that drive tumor growth and recurrence, authors completely failed to even touch the later aspect. It is recommended that authors dedicate a paragraph to the topic of the effectiveness of the proposed therapies towards cancer stem cells, or at least mention in relevant places in the text the effects of the discussed biologics and small molecules on the toxicity towards cancer stem cells, where such knowledge is available. Authors may be inspired by recent publications (doi: 10.1186/s12935-020-01718-6, doi: 10.1016/j.ejphar.2020.173202, doi: 10.3390/ijms21072263)

2.)

The reviewer is thankful to the authors for summarizing in tables 1 & 2 the critical parts of the message, however both tables should be expanded by adding a middle column that briefly explains the nature of the compound, and or the precise mode of action in a couple of phrases, if possible. This would greatly enhance the usefulness and likely appreciation by the readership.

Minor:

- it is unclear why “Hematopoietic Stem Cells” were capitalized in the end of the sentence at page 6?
- please take care of other occasional minor dramatic and orthography mistakes.

Author Response

Q.1 “It is recommended that authors dedicate a paragraph to the topic of the effectiveness of the proposed therapies towards cancer stem cells, or at least mention in relevant places in the text the effects of the discussed biologics and small molecules on the toxicity towards cancer stem cells, where such knowledge is available. Authors may be inspired by recent publications (doi: 10.1186/s12935-020-01718-6, doi:  10.1016/j.ejphar.2020.173202, doi: 10.3390/ijms21072263)”.

Response: We thank the Reviewer for this important suggestion. Accordingly, in the text, we added an entire paragraph and some sentences about cancer stem cells characteristics and their role in tumor progression. We thank the Reviewer also for the publications suggested.

Q.2 The reviewer is thankful to the authors for summarizing in tables 1 & 2 the critical parts of the message, however both tables should be expanded by adding a middle column that briefly explains the nature of the compound, and or the precise mode of action in a couple of phrases, if possible. This would greatly enhance the usefulness and likely appreciation by the readership.

Response: We accepted the comment and modified the Tables which now include the class of compounds, the number and status of the clinical trial reported, the type of tumor and the relative references when available, as suggested also by Reviewer #3. The addition of some phrases in a column of the tables we thought was not graphycally possible; instead, we added more details in the text and in the figures.

Q.3 “it is unclear why “Hematopoietic Stem Cells” were capitalized in the end of the sentence at page 6?”

Response: We apologize for this inaccuracy that was eliminated and the words are now written in small letters.

Q.4 “please take care of other occasional minor dramatic and orthography mistakes.”

Response: The entire manuscript has been revised for any typos or orthography mistakes that were removed.

Reviewer 3 Report

See the attached document.

Author Response

Q.1 “Tables: the tables with the different drugs in development to act on the CXCR4 and CD47 receptors should be more complete than the current ones. These tables should indicate the different clinical trials in development, as well as whether it is possible in many of them the results of the trials that are already available and the types of tumors on which they have been carried out.

Response: We accepted the suggestion and modified the Tables. They now include the class of compounds, the number and status of the clinical trial reported, the type of tumor and the relative references when available, as suggested also by Reviewer #2. The results of each clinical trial have been reported in text, when mentioned.

Q.2 “Figures: at least two figures must be made for the article that explain the recipients to be studied with their pathways. If this point is not clarified it is very difficult from my point of view to accept the article because it is too dense and complex to understand by readers.”

Response: We would like to thank the Reviewer for this comment. In the revised version of the manuscript, we added two cartoon-figures which illustrate the pathways activated by the two receptors. We hope that they will make the article easier to follow and less complex to understand by a wide readership.

Q.3 “Title: as indicated above, the title should reflect that this is a review of the literature so that this helps the reader to correctly select the article he wants to study.”

Response: We accepted the comment and modified the title by stating that this is an overview of the literature on the CXCR4 and CD47 receptors antagonists.

Q.4 “Introduction: specify in the second paragraph that the treatment of tumors is based on both genomic and clinical data, and that cancer treatments should be individualized according to these two characteristics”.

Response: We thank the Reviewer for this comment and amended the sentence in the second paragraph of the Introduction section, as suggested.

Q.5 “Introduction: in the fourth paragraph when talking specifically about CXCR4, indicate about which type of tumors has greater importance, so that the reader can understand the importance of the study (for example, colon or breast).”

Response: A sentence stating what suggested has now been included in the fourth paragraph of the Introduction.

Q.6 “CXCR4 Antagonist: it could go into more detail and give more results on the responses in clinical trials, especially in the most important tumors such as breast cancer, for example”.

Response: We thank the Reviewer for his/her comment that we took into consideration. In the revised version, the most important tumors targeted by each CXCR4 antagonist have been indicated.

Q.7 “POL6326: Specify the number of the clinical trial, as well as in the phase 1b clinical trial of breast cancer specify whether it is triple-negative tumors or only those that are HER2 negative”.

Response: For this specific compound, the numbers of the clinical trials have been added, as well as the type of tumor. 

Q.8 “Conclusions: in the first sentence of the conclusions, it would be correct to change the word "dismal" to "uncertain".

Response: The correction has been made.

Q.9 “Conclusions: The phrase "In this regard, we focused..." I think it could be eliminated because it does not contribute anything in the conclusions”.

Response: The sentence has been rephrased in the new version of the manuscript.

Round 2

Reviewer 2 Report

May be accepted if approved by the staff-editor.

Author Response

Response to Reviewer #2

We would like to thank the Reviewer for Her/His work.

We have revised the manuscript for the English language and style.

Author Response

Response to Reviewer #3

We would like to thank the Reviewer for Her/His work and for the

suggestions made to improve the quality of the manuscript.